# p75NTR Modulation Reduces Oxidative Stress and the Expression of Pro-Inflammatory Mediators in a Cell Model of Rett Syndrome

**DOI:** 10.3390/biomedicines12112624

**Published:** 2024-11-16

**Authors:** Michela Varone, Giuseppe Scavo, Mayra Colardo, Noemi Martella, Daniele Pensabene, Emanuele Bisesto, Andrea Del Busso, Marco Segatto

**Affiliations:** 1Department of Biosciences and Territory, University of Molise, Contrada Fonte Lappone, 86090 Pesche, Italy; m.varone@studenti.unimol.it (M.V.); giuseppe.scavo@unimol.it (G.S.); mayra.colardo@unimol.it (M.C.); n.martella@studenti.unimol.it (N.M.); d.pensabene@studenti.unimol.it (D.P.); e.bisesto@studenti.unimol.it (E.B.); a.delbusso@studenti.unimol.it (A.D.B.); 2Department of Science, University Roma Tre, 00146 Rome, Italy

**Keywords:** Rett syndrome, neurodevelopmental disorder, oxidative damage, redox imbalance, inflammation, neurotrophins, LM11A-31

## Abstract

**Background:** Rett syndrome (RTT) is an early-onset neurological disorder primarily affecting females, leading to severe cognitive and physical disabilities. Recent studies indicate that an imbalance of redox homeostasis and exacerbated inflammatory responses are key players in the clinical manifestations of the disease. Emerging evidence highlights that the p75 neurotrophin receptor (p75NTR) is implicated in the regulation of oxidative stress (OS) and inflammation. Thus, this study is aimed at investigating the effects of p75NTR modulation by LM11A-31 on fibroblasts derived from RTT donors. **Methods:** RTT cells were treated with 0.1 µM of LM11A-31 for 24 h, and results were obtained using qPCR, immunofluorescence, ELISA, and Western blot techniques. **Results:** Our findings demonstrate that LM11A-31 reduces OS markers in RTT fibroblasts. Specifically, p75NTR modulation by LM11A-31 restores protein glutathionylation and reduces the expression of the pro-oxidant enzyme NOX4. Additionally, LM11A-31 significantly decreases the expression of the pro-inflammatory mediators interleukin-6 and interleukin-8. Additionally, LM11A-31 normalizes the expression levels of transcription factors involved in the regulation of the antioxidant response and inflammation. **Conclusions:** Collectively, these data suggest that p75NTR modulation may represent an effective therapeutic target to improve redox balance and reduce inflammation in RTT.

## 1. Introduction

Rett syndrome (RTT, OMIM 312750) is an early onset, progressive, severe neurodevelopmental disorder [1]. It is linked to the X chromosome and affects females mainly, with a frequency of 1:15,000 live births [1,2]. In 95% of cases, the classical form of RTT is associated with mutations in the *mecp2* gene [3], codifying for the methyl-CpG-binding protein 2 (MeCP2), which has the role of an epigenetic modulator. Most patients with RTT develop normally up to 6–18 months of age [1]. Subsequently, they present with a typical clinical picture that includes four stages of neurological regression, summarized by the loss of cognitive, social, and motor skills acquired in early life [4,5]. 

RTT is no longer considered an exclusively neurological disease but a multisystem syndrome that affects the brain and several other tissues/organs. Indeed, increasing evidence suggests that the systemic nature of RTT could be the clinical consequence of two synergistic conditions: oxidative stress (OS) and subclinical inflammation [6,7,8]. 

To date, numerous studies have shown that RTT is associated with redox alterations. Accordingly, it has been shown that oxidative damage to macromolecules is present in RTT mouse models even before the earliest typical symptoms become evident [6,7,9,10,11,12,13,14]. Additionally, in patients, indicators of exacerbated membrane damage are evident from the early stages of the disease and increase as it progresses [15]. To date, the causes of redox imbalance in RTT are less clear, but mitochondrial dysfunction is considered a crucial event [16,17,18,19]. Furthermore, xanthine oxidase (XO) and NADPH oxidase (NOX), non-mitochondrial sites of ROS production, can also be responsible for redox alterations in the pathology. For instance, an increased NOX activity is reported in the fibroblasts of RTT patients [13]. 

Over the years, the relationship between OS and inflammation has been extensively described by the scientific community and is now universally accepted. Inflammation itself, in fact, represents a source of OS. During the inflammatory process, oxidant molecules are generated from the activity of cyclooxygenase, lipoxygenase, and NOX [20,21]. Therefore, it appears credible that the chronic and systemic OS documented in RTT could induce a state of “mild subclinical chronic inflammation” [8]. Recently, numerous findings have been collected in support of these hypotheses and the connection between the inflammatory state and OS in RTT. An altered cytokine profile has been documented in RTT patients, showing an exacerbation of the T-helper type 2 (Th2) cytokine-mediated response with significantly elevated levels of interleukin-5 (IL-5) and -6 (IL-6) compared to controls [22]. Supporting these data, another study found increased levels of IL-8, IL-9, and IL-13 in RTT patients compared to healthy subjects [8].

Numerous studies also highlighted the involvement of neurotrophins in RTT pathology. Deregulation of neurotrophin signaling appears to be involved in the onset and severity of disease symptomatology in RTT animal models [23,24,25]. Neurotrophins exert their biological functions by binding high-affinity Trk receptors and the low-affinity p75 neurotrophin receptor (p75NTR). While the involvement of the high-affinity receptors, such as TrkA and TrkB, has already been described in RTT [25,26], the putative contribution of p75NTR in RTT has not yet been examined. The p75NTR receptor is recognized as a receptor for all neurotrophins. Besides its well-known role in regulating cell growth and differentiation [27], numerous studies also indicate that this receptor may govern pathways involved in the regulation of OS and inflammation [28].

The physiopathological involvement of p75NTR has led to the identification of peptides and small non-peptidic molecules capable of modulating its activity. Among these, LM11A-31 is a small ligand specifically designed to bind to p75NTR within the NGF *loop1* binding domain [29]. LM11A-31 acts neither strictly as an agonist nor as an antagonist, allowing for pro-survival signaling induced by phosphoinositide 3-kinase (PI3K)/protein kinase B (PKB) while suppressing degenerative signaling [29].

There are no definitive cures for RTT, and the available treatments aim to mitigate and control its symptoms. At the same time, as reported above, the role of the p75NTR receptor has not yet been thoroughly investigated in this pathology, and recent evidence regarding its participation in redox balance offers hope for its involvement in the pathophysiology of RTT.

Therefore, the aim of this study was to evaluate whether the modulation of p75NTR by LM11A-31 could counteract the OS and inflammation in RTT. To this end, primary cell cultures of fibroblasts derived from RTT patients and healthy donors were used as experimental models. Fibroblasts from RTT individuals can be considered valuable in vitro models of the disease, as they adequately mirror the systemic OS and inflammation found in RTT patients [30].

## 2. Materials and Methods

### 2.1. Cell Culture and Treatments

Fibroblasts were obtained via skin biopsy from two healthy female donors (mean age 19 years), used as a healthy control (HC) group, and from two age-matched female donors diagnosed with classical RTT (mean age 20 years). These cell cultures were employed in our previous studies; details about subject selection, ethic statements, and inclusion/exclusion criteria have already been reported in Segatto et al., 2014 [31]. Cells were cultured at 5% CO_2_ in Dulbecco’s Modified Eagle Medium (DMEM) at high glucose (Merck Life Science, Milan, Italy, D6429), containing 4.5 g/L of glucose, 2 mM L of glutamine, and 100 mM of sodium pyruvate. The culture medium was enriched with 15% (*v*/*v*) fetal bovine serum (FBS) (Sigma Aldrich, Milan, Italy, F7524) and a 1% antibiotic solution composed of penicillin/streptomycin (PAN Biotech, Aidenbach, Germany, P06-07100). A 10% FBS was used for the experiments. Fibroblasts were seeded and, 24 h later, treated with LM11A-31 (Sigma Aldrich, Milan, Italy, SML00664) at a concentration of 0.1 μM for 24 h. The compound was dissolved in dimethyl sulfoxide (DMSO) (Sigma Aldrich, Milan, Italy, D5879). LM11A-31 was first dissolved in DMSO to obtain a stock solution of 100 mM, which was then aliquoted and stored at −80 °C. Immediately before use, aliquots of LM11A-31 in DMSO were thawed and diluted 1:1000 in culture medium (DMEM + 10% FBS) to obtain a working solution of 100 μM of LM11A-31. The working solution was then added to the cultured cells (1:1000 dilution) to give a final concentration of 100 nM of LM11A-31, a dose generally employed in cell culture studies [32,33,34]. As a result, the DMSO concentration in the cell culture was 0.0001%. This negligible concentration was well below the concentration range commonly used to dissolve drugs in cell culture experiments. To make sure that all experimental groups were handled under the same conditions, in all experiments involving LM11A-31 treatment, HC (healthy control) and RTT (pathological control) cells received 0.0001% DMSO as vehicles in the culture medium. Notably, we experimentally confirmed that 0.0001% DMSO did not influence the parameters of OS and inflammation evaluated in this work (Appendix A). All experiments were conducted at 60–70% cell confluency and between 10 and 15 passages.

### 2.2. Cell Lysis and Western Blotting Analysis

Cell lysis and protein extracts were prepared according to Colardo et al., 2022, with modifications [35]. Briefly, protein extracts were obtained by lysing the cells with a lysis buffer composed of 10 mM of HEPES (pH 8), 10 mM of KCl, 0.1% NP-40, 0.5 mM of DTT, and enriched with a protease and phosphatase inhibitor cocktail. Lysates were then centrifuged for 7 min at 12,000 RCF at 4 °C. The supernatants were collected, and the protein concentrations were determined using the Lowry protein assay (Bio-Rad Laboratories, Milan, Italy). Laemmli Buffer 5X (Tris-HCl 315 mM, pH 6.8; 2.5% β-mercaptoethanol; 50% glycerol; 10% sodium dodecyl sulfate; and 0.5% Bromophenol Blue) was added to supernatants, and the samples were boiled at 95 °C for 5 min. The protein extracts were separated on an SDS-PAGE, and proteins were transferred onto nitrocellulose (GE Healthcare, Life Sciences, Little Chalfont, Buckinghamshire, UK) using the Trans-Blot Turbo transfer system (Bio-Rad Laboratories, Milan, Italy). After blocking with 5% fat-free milk powder in Tris-buffered saline and 0.1% Tween-20, the membranes were probed for 24 h at 4 °C with the primary antibodies diluted in 5% Milk in a PBS-Tween solution. The day after, the membranes were washed with PBS-Tween to remove the unbound antibodies and subsequently probed with the secondary antibodies conjugated to horseradish peroxidase, diluted in 5% Milk in PBS-Tween. For the detection of the chemical signal, the protein antibody immunocomplexes were detected using an ECL chemiluminescence system (GE Healthcare, Life Sciences, Little Chalfont, Buckinghamshire, UK), and chemiluminescence was recorded using a ChemiDoc MP system (Bio-Rad Laboratories, Milan, Italy). The following primary antibodies were used: anti-SOD1 (Santa Cruz Biotechnology, Dallas, TX, USA, sc-271014), anti-SOD2 (Santa Cruz Biotechnology, Dallas, TX, USA, sc-137254), anti-catalase (Santa Cruz Biotechnology, Dallas, TX, USA, sc-271803), anti-TrxR1 (Santa Cruz Biotechnology, Dallas, TX, USA, sc-28321), anti-GPx-1 (Abcam, Cambridge, UK, ab22604), and anti-Glyceraldehyde-3-phosphate dehydrogenase (GAPDH) (Santa Cruz Biotechnology, Dallas, TX, USA sc-32233). The detection was achieved using horseradish peroxidase-conjugated secondary antibodies anti-mouse (Bio-Rad Laboratories, Milan, Italy, #1706516) and anti-rabbit (Bio-Rad Laboratories, Milan, Italy, #1706515). The chemical signal was detected using clarity ECL Western blotting (Bio-Rad Laboratories, Milan, Italy, No. 1705061), and chemiluminescence was measured with a ChemiDoc MP system (Bio-Rad Laboratories, Milan, Italy). The acquired images were subject to densitometric analysis using ImageJ software version 1.53k (National Institutes of Health, Bethesda, MD, USA) for Windows.

### 2.3. RNA Extraction and Real-Time PCR

qRT-PCR was performed as previously described, with modifications [35,36]. Total RNA was extracted from fibroblasts using the Trizol reagent (Sigma Aldrich, Milan, Italy, T9424) in accordance with the manufacturer’s guidelines. Post-extraction, the RNA was treated with DNAse (Ambion/Life Technologies, Paisley, UK) to eliminate any residual DNA. Subsequently, RNA purification was executed using the RNA Clean-up Kit (Zymo, Irvine, CA, USA), and the purified RNA was reverse transcribed to cDNA using the High-Capacity cDNA Reverse Transcription Kit (Applied Bio-System), which was then subjected to qPCR analysis. Primers utilized in qRT-PCR were as follows: *ngf*, 5′-ACCCGCAACATTACTGTGGACC-3′ (forward) and 5′-GACCTCGAAGTCCAGATCCTGA-3′ (reverse); *ngfr*, 5′-CCTCATCCCTGTCTATTGCTCC-3′ (forward) and 5′-GTTGGCTCCTTGCTTGTTCTGC-3′ (reverse). qPCR was conducted in triplicate utilizing the SYBR Green IQ reagent (Bio-Rad Laboratories, Milan, Italy).

### 2.4. Immunofluorescence Staining and Confocal Analysis

Cells were treated as previously reported and fixed in a 4% PFA solution in PBS for 10 min, followed by permeabilization in PBS-Triton 0.1% for 5 min at room temperature. Cells were then blocked with Bovine Serum Albumin (BSA) dissolved in PBS-Triton 0.1% for 30 min and then incubated with the primary antibody overnight at 4 °C. The following primary antibodies were used in the immunocytochemistry experiments: anti-NGF (Santa Cruz Biotechnology, Dallas, TX, USA, sc-365944), anti-p75NTR (Santa Cruz Biotechnology, Dallas, TX, USA, sc-271708), anti-8-OHdG (Santa Cruz Biotechnology, Dallas, TX, USA, sc-66036), anti-4-HNE (Abcam, Cambridge, UK, ab46545), anti-GSH (Abcam, Cambridge, UK, ab19534), anti-NOX4 (Santa Cruz Biotechnology, Dallas, TX, USA, sc-518092), anti-p22^phox^ (Santa Cruz Biotechnology, Dallas, TX, USA, sc-130551), anti-IL-6 (Santa Cruz Biotechnology, Dallas, TX, USA, sc-129128), anti-IL-8 (Santa Cruz Biotechnology, Dallas, TX, USA, sc-8427), anti-PPAR-α (Abcam, Cambridge, UK, ab314112), anti-PPAR-β/δ (Abcam, Cambridge, UK, ab23673), anti-PPAR-γ (Santa Cruz Biotechnology, Dallas, TX, USA, sc-7273), anti-PGC1α (Santa Cruz Biotechnology, Dallas, TX, USA, sc-13067), anti-Sirt1(Santa Cruz Biotechnology, Dallas, TX, USA, sc-74465), and anti-Nrf2 (Santa Cruz Biotechnology, Dallas, TX, USA, sc-365949). All primary antibodies employed in this study are commercially available and were selected based on their published validation. The anti-4-HNE, anti-8-OHdG, and anti-Nrf2 antibodies were subjected to further validation in our laboratory on different human cell lines (HepG2, SH-SY5Y, and fibroblasts) treated with OS inducers (300 µM of H_2_O_2_ and 0.1 µM of rotenone) as a positive control. The anti-IL-6 and anti-IL-8 antibodies were validated using LPS-stimulated THP-1 cells as a positive control. The anti-NGF and anti-p75NTR antibodies were validated using NGF (BLP-N240, Alomone labs, Jerusalem, Israel) and p75NTR (BLP-NT011 Alomone labs, Jerusalem, Israel) blocking peptides.

Following this, the cells were incubated in the dark for 1 h at room temperature with goat anti-mouse secondary antibody Alexa Fluor 555 (Thermo Fisher Scientific, Waltham, MA, USA, A28180) and goat anti-rabbit secondary antibody Alexa Fluor 488 (Thermo Fisher Scientific, Waltham, MA, USA, A27034). After nuclear staining with DAPI (Thermo Fisher Scientific, Waltham, MA, USA), the coverslips were mounted with Fluoroshield mounting medium (Merck Life Science, Milan, Italy, F6182) and then analyzed using a confocal microscope (TCS SP8; Leica, Wetzlar, Germany) equipped with a 40 × 1.40–0.60 NA HCX Plan Apo oil BL objective. Images were acquired and analyzed using Leica Application Suite X (LAS X) software (version 3.5.5).

Signal quantification was calculated as the mean fluorescence intensity per cell area using ImageJ v1.54d software (National Institutes of Health, Bethesda, MD, USA) for Windows 10, according to the previously reported procedure [37]. To ensure the absence of operator biases, the results concerning immunofluorescence quantification were additionally confirmed through the automated analysis of the same images with the software Q-IF [37].

Histograms were chosen for the visualization of quantitative information on immunopositivity. The dots dispersed around the SD represent individual values, each derived from the average fluorescence of the cells analyzed in a single image (each image is from a different experiment).

### 2.5. ELISA

ELISA assay kits for IL-6 (Thermo Fisher Scientific, Waltham, MA, USA, # EH2IL6) and IL-8 (Thermo Fisher Scientific, Waltham, MA, USA, # KHC0081) were used to quantify the concentration of interleukins released in the culture medium, according to the manufacturer’s instructions.

### 2.6. Statistical Analysis

All the results obtained from the statistical analyses are expressed as means ± SDs (standard deviations). An unpaired student’s *t*-test was utilized to compare two experimental groups, while a one-way analysis of variance (ANOVA) followed by Tukey’s post hoc test was employed to compare three or more groups. *p*-values < 0.05 define a significant difference between experimental groups. The statistical analysis and editing of the graphs shown were performed using GraphPad Prism version 8.0 software (GraphPad, La Jolla, CA, USA) for Windows.

## 3. Results

### 3.1. NGF and p75NTR Expression Is Reduced in RTT Fibroblasts

First, we examined the expression of NGF in HC and RTT fibroblasts. According to previous reports [38], mRNA and immunofluorescence analysis showed that NGF is produced by fibroblasts. Interestingly, the expression of this neurotrophin was markedly decreased in RTT cells at both the transcript and protein levels (Figure 1A,B). Fibroblasts not only produce NGF but also respond to this neurotrophin by expressing p75NTR. Notably, we found that the relative abundance of p75NTR was dramatically decreased in RTT fibroblasts at both mRNA and protein levels, suggesting a downregulation of the NGF/p75NTR signaling pathway (Figure 1C,D).

### 3.2. Modulation of p75NTR Reduces Oxidative Stress in Fibroblasts Derived from Patients Affected by RTT

Once we assessed the imbalance in the expression of NGF and p75NTR, we evaluated whether p75NTR modulation by LM11A-31 would counteract redox damage in RTT cells. For this purpose, the presence of some specific markers of OS was evaluated. As shown by immunofluorescence analysis, the relative abundance of 8-hydroxy-2’-deoxyguanosine (8-OHdG), a marker of oxidative damage to nucleic acids, was significantly higher in RTT fibroblasts compared to HCs. However, the exaggerated immunoreactivity was effectively reduced by LM11A-31 treatment, resulting in restoration to near-baseline levels of HC cells (Figure 2A). Similar results were obtained by evaluating the immunoreactivity of 4-HNE, a product of lipid peroxidation that is widely used as an OS marker [39]. In particular, the increased 4-HNE immunoreactivity observed in RTT fibroblasts was significantly counteracted by LM11A-31 administration (Figure 2B). The effectiveness of LM11A-31 in mitigating OS was corroborated through the examination of 8-OHdG in cultured fibroblasts derived from other HC and RTT subjects (Appendix A). This evidence substantiates the assertion that p75NTR modulation governs redox homeostasis irrespective of the individual’s background. Overall, these data demonstrate that pharmacological manipulation of p75NTR attenuates the redox imbalance in RTT.

### 3.3. Dysregulation of Redox Homeostasis in RTT Fibroblasts Is Partially Reversed by p75NTR Modulation

The results previously collected led us to investigate the expression of the enzymes involved in redox homeostasis, particularly those involved in the antioxidant response. While no changes were observed in the expression of superoxide dismutase 1 (SOD1) (Figure 3A), superoxide dismutase 2 (SOD2), catalase, thioredoxin reductase 1 (TrxR1), and glutathione peroxidase 1 (Gpx1) were found to be dramatically downregulated in RTT fibroblasts. Notably, these alterations were not counteracted by LM11A-31 treatment (Figure 3B–E).

The extent of protein glutathionylation was next evaluated through immunofluorescence. This post-translational modification determines the attachment of glutathione moieties to proteins, leading to the regulation of redox-based signaling events and serving as protective mechanisms against oxidative damage [40]. Our results showed that immunostaining for GSH-protein complexes is lower in RTT-1 fibroblasts compared to HC-1 fibroblasts. Importantly, pharmacological p75NTR modulation by LM11A-31 significantly attenuated the pathological reduction of protein glutathionylation in RTT fibroblasts, being restored to HC-1 levels (Figure 3F).

The attention was then focused on the NADPH oxidase 4 complex, one of the main constitutively active pro-oxidant systems of the cell. NADPH oxidase 4 is a multimeric enzyme belonging to the oxidoreductase class, which predominantly catalyzes the formation of hydrogen peroxide [41]. The active complex is composed of NOX4 and p22^phox^ subunits. The expression of this enzymatic complex has already been reported in fibroblasts [42]; specifically, NOX4 is a gp91^phox^ homolog highly expressed in fibroblasts and is frequently modulated in different pathological conditions [43]. Confocal analysis showed that NOX4-associated immunopositivity is greater in RTT cells compared to that found in HC cells. However, this alteration was completely prevented by LM11A-31 administration (Figure 4A). Conversely, p22^phox^ protein levels did not show significant variations in the three experimental groups considered in the study (Figure 4B). Collectively, these results indicate that p75NTR modulation by LM11A-31 mitigates oxidative damage by properly restoring protein glutathionylation and NOX4 expression in RTT cells.

### 3.4. p75NTR Modulation Hinders the Increase in Inflammatory Mediators in RTT Cells

Oxidative processes are frequently associated with inflammatory contexts, so much so that the combination of the two pathological events observed in RTT has led to the definition of the term *Oxinflammation* [8]. This genetic disease, in fact, is characterized by a parallel alteration of systemic inflammation and a compromised redox balance. In this work, the pro-inflammatory environment was assessed by analyzing the expression of IL-6 and IL-8, two pro-inflammatory mediators strictly associated with the induction of OS [44,45]. Notably, changes in IL-6 and IL-8 have already been reported in RTT individuals [46]. The confocal analysis and ELISA assay showed that LM11A-31 administration normalized the expression of IL-6 in RTT-1 cells and culture medium, respectively, to HC-1 levels (Figure 5A,B). LM11A-31 also reduced IL-6 levels in fibroblasts derived from another RTT individual (Appendix A). Moreover, IL-8 levels were markedly raised in RTT-1 cells and the corresponding conditioned medium. However, this elevation was substantially attenuated by LM11A-31 (Figure 5C,D). Collectively, these results suggest that RTT cells produce abnormal levels of proinflammatory interleukins, reflecting the systemic inflammation that occurs in the pathology. Remarkably, the administration of LM11A-31 counteracts the increase in inflammatory mediators.

### 3.5. p75NTR Modulation by LM11A-31 Normalizes the Expression of Transcriptional Regulators Involved in Anti-Inflammatory Response and Redox Homeostasis in RTT Fibroblasts

OS and inflammation often lead to the regulation of gene transcription by affecting the expression/activation of a variety of transcription factors, such as peroxisome proliferator-activated receptors (PPARs). In particular, these transcriptional regulators are involved in the modulation of both antioxidant and anti-inflammatory responses [47]. Immunofluorescence analysis performed in this study provided evidence that PPARα and PPARγ expression, whose localization is predominantly nuclear, is higher in RTT-1 fibroblasts when compared to HC-1 cells. However, these alterations were significantly normalized to HC-1 levels following treatment with the p75NTR modulator (Figure 6A,C). In contrast, the expression of PPARβ/δ appears to be both nuclear and cytoplasmic and does not show any statistically significant variation in the three different experimental groups (Figure 6B).

PPAR-mediated transcriptional control is also influenced by peroxisome proliferator gamma coactivator 1 (PGC1α) [48]. Our results revealed that PGC1α is expressed in HC fibroblasts, showing diffuse staining but mostly restricted to the nuclear level. Interestingly, its immunoreactivity was markedly reduced in RTT fibroblasts but reversed upon LM11A-31 administration (Figure 7A). Another key protein involved in OS and inflammation is sirtuin 1 (Sirt1) [49]. The immunopositivity associated with Sirt1, mainly confined to the nuclear level, tends to be higher in RTT fibroblasts when compared to HC cells, although this increase was not statistically significant. LM11A-31 treatment was not able to restore the basal levels observed in HC fibroblasts (Figure 7B).

Next, we evaluated the expression of the nuclear transcription factor erythroid-2 (Nrf2). Nrf2 is a central regulator not only of the antioxidant response but also of the expression of hundreds of genes that control many processes, including immune and inflammatory responses, tissue remodeling, and fibrosis [50]. Nrf2 activity is tightly regulated by a complex network of transcriptional and post-translational mechanisms, allowing it to orchestrate cell response and adaptation to various stressors to maintain cellular homeostasis [51]. In particular, OS stabilizes Nrf2, which then translocates to the nucleus and mediates the transcription of genes involved in the antioxidant response through a ROS-induced feedback response [52]. The immunopositivity associated with Nrf2, which was mainly localized at the nuclear level, was more pronounced in RTT fibroblasts than in HC cells. Notably, LM11A-31 treatment restored Nrf2 immunoreactivity to control levels (Figure 7C).

## 4. Discussion

It is becoming increasingly clear that the imbalance in redox homeostasis and the exacerbated increase in the inflammatory response contribute to the multiple clinical manifestations of RTT [8,53]. Specifically, the altered balance in redox homeostasis leads to a subclinical inflammatory state that contributes to the accumulation of ROS, creating a vicious cycle that induces a chronic pro-inflammatory state known as *Oxinflammation*. Experimental evidence demonstrates that *Oxinflammation* is not just a consequence characterizing this pathology but a true pathological mechanism contributing to the disease progression [8].

In this study, we investigated the effects of p75NTR modulation by the small molecule LM11A-31 on the redox balance in RTT. Notably, we found that both NGF and p75NTR expression are markedly decreased in RTT fibroblasts, supporting the notion that NGF/p75NTR pathway is altered in this pathology. In this context, our results corroborate previous findings, showing that NGF levels were reduced in the brain and blood of RTT patients [25].

Compelling evidence highlights that p75NTR activity is particularly involved in counteracting OS and inflammation in diverse physiopathological conditions [54,55].

According to the literature data, our results sustain that OS is particularly elevated in RTT fibroblasts. Specifically, the increase in 8-OHdG immunoreactivity supports the previous findings collected in both human and animal models of RTT [56]. Similarly, OS in RTT fibroblasts is particularly evidenced by the increased levels of the lipid peroxidation marker 4-HNE. These results are consistent with those obtained from plasma samples of RTT patients, which show a significant increase in malondialdehyde (MDA) concentration [9]. Sierra and colleagues hypothesized that the elevated levels of this lipid peroxidation end product represent peroxidative damage to biological membranes, which may contribute to the typical symptoms of the disease, including progressive dementia, impaired motor functions, behavioral changes, and seizures [9]. In recent years, numerous data obtained from studies on patients and animal models of the disease have shown that RTT is characterized not only by oxidative damage to lipids, resulting in the production of 4-HNE, but also by protein damage caused by the formation of 4-HNE adducts (4-HNE-PA) [10,13,57]. The overwhelming evidence of the pathogenic role of redox imbalance in RTT is given by the considerable phenotypic improvement of the MeCP2-*null* mice, correlated with the decrease in levels of the brain OS markers after the re-expression of MeCP2 [10]. Therefore, increased ROS and subsequent redox imbalance can be considered factors driving the progression of RTT pathophysiology [6,7].

In this context, data collected in our study show that pharmacological targeting of p75NTR by LM11A-31 hinders oxidative damage, suggesting that alterations in the p75NTR signaling pathway may significantly contribute to derangements of the redox balance in RTT.

The expression of SOD2, catalase, TrxR1, and Gpx1 was extremely low in RTT fibroblasts, supporting the establishment of a pro-oxidant environment. These data corroborate the results obtained in a previous study, which observed reduced activity and/or expression of these enzymes in similar experimental models of the disease [13]. Unfortunately, LM11A-31 was not able to improve the observed dysregulations.

At the same time, glutathionylated proteins were significantly reduced in RTT fibroblasts compared to HC fibroblasts, supporting the previously observed depletion of the GSH pool in post-mortembrain samples from RTT individuals [58]. Interestingly, we noted that GSH-protein complexes were particularly pronounced at nuclear levels in healthy cells if compared to RTT fibroblasts. The literature data demonstrate that GSH accumulates in the nucleus at the beginning of the G1 phase of cell proliferation and may, therefore, play a key role in preserving the redox state of the nucleus during the cell cycle. Indeed, the presence of GSH within the nucleus could ensure a highly reducing environment that protects genomic DNA from oxidative damage, particularly in the event of nuclear envelope breakdown during the G1 and G2/M phases [59]. Additionally, GSH can regulate chromatin conformation and the accessibility of the repair mechanism [60]. However, many questions remain unanswered regarding how GSH is sequestered in the nucleus and its function in genetic and epigenetic processes. In this regard, it will be interesting to delve deeper into the biological significance of GSH depletion in RTT cells and its possible contribution to determining the pathological phenotype. Notably, LM11A-31 significantly increased the number of GSH-protein complexes, suggesting that the attenuation of OS mediated by p75NTR modulation involves the regulation of GSH metabolism. The role of p75NTR has already been associated with the modulation of the antioxidant machinery; p75NTR-negative cells are less resilient to oxidative damage in PC12 cells, supporting the notion that reduced p75NTR expression may also affect redox homeostasis in RTT cells. Notably, the intracellular domain of p75NTR is an essential prerequisite for regulating glutathione oxidation and protecting PC12 cells against oxidative stress [61]. Despite this evidence, the molecular pathways involved in the antioxidant effects of p75NTR need to be further investigated. The intricate nature of the signal transduction cascades associated with p75NTR presents a fascinating challenge to gain deeper insights into its biological functions.

Besides changes in the cellular antioxidant system, we found that the pro-oxidant NADPH oxidase 4 complex is also altered in RTT cells and may contribute to the oxidative damage observed in the pathology. Specifically, while no significant differences were observed for the p22^phox^ subunit, the expression levels of the NOX4 subunit are significantly higher in RTT cells compared to HC, consistent with the increased NOX4 enzymatic activity already reported in the literature [13]. Overall, these results support the idea that non-mitochondrial sites of ROS production also contribute to the OS observed in RTT. In this context, it is important to emphasize that the modulation of p75NTR exerted by LM11A-31 efficiently normalizes the basal levels of NOX4. To date, the molecular mechanisms linking p75NTR activity and the modulation of NADPH oxidase subunits are still unexplored and deserve further investigation. Nevertheless, elegant work has demonstrated that p75NTR loss markedly diminishes the expression of Siah2, a ubiquitin ligase responsible for HIF-1α degradation. Consequently, the absence of p75NTR promotes the stabilization of HIF-1 [62]. As the *nox4* gene is under the transcriptional control of HIF-1α [63], it is possible to speculate that the reduction of p75NTR observed in RTT fibroblasts facilitates the upregulation of the NOX4 subunit through the contribution of HIF-1α.

Oxidative processes are frequently associated with inflammatory contexts. In this regard, we found exaggerated expression of IL-6 and IL-8 in RTT fibroblasts. Accordingly, previous studies detected high concentrations of pro-inflammatory cytokines in blood samples from patients with RTT [22]. Moreover, the predictive value of salivary tests for assessing immuno-inflammatory alterations in the disease has been recently demonstrated, highlighting an increase in IL-1β, IL-6, and IL-8 in saliva samples from RTT patients and a strong correlation between cytokine concentration and symptom severity [46]. Furthermore, another study found increased levels of IL-8, IL-9, and IL-13 in RTT patients compared to controls [8]. In RTT patients, a condition of chronic intermittent hypoxia associated with OS has been described, along with altered pulmonary gas exchange [6]. This condition can activate a series of transcription factors, such as nuclear factor Kappa B (NF-kB), which induces inflammatory mediators (primarily IL-8), thus contributing to the dysregulation of pro-inflammatory cytokines observed in RTT [8]. Indeed, upregulated NF-kB signaling has been documented in the brain cortex of mouse models with MeCP2 loss of function [64]. Furthermore, the survival of MeCP2-null mice was extended following genetic suppression of NF-kB [64]. The absence of MeCP2 also increases the expression of pro-inflammatory cytokines such as IL-6 and tumor necrosis factor- α (TNF-α) [65]. Our data demonstrate that the exacerbated expression of cytokines is drastically reduced by LM11A-31 treatment, highlighting the significant impact of p75NTR modulation in regulating the inflammatory response. Indeed, numerous scientific studies report a significant attenuation of pro-inflammatory pathways upon p75NTR manipulation by LM11A-31 [66,67,68]. Although the downstream pathways of p75NTR leading to modulation of inflammation are still unclear, it has been shown that this receptor can recruit signaling proteins such as interleukin receptor-associated kinase (IRAK), which are generally recruited by interleukin-1 receptor (IL1R) or toll-like receptors (TLRs) to activate NF-kB and promote a pro-inflammatory response [69]. Thus, it has been postulated that p75NTR may sequester IRAK, which is no longer available to compete with receptors responsible for initiating the inflammatory cascade [54]. Supporting this hypothesis, other findings show that p75NTR modulation by LM11A-31 suppresses NF-kB activation in diverse physiopathological contexts [70,71].

Genes involved in OS and inflammatory response are regulated by various transcription factors. Our results show strong nuclear expression of PPARα, PPARγ, and Nrf2 in RTT fibroblasts, which were effectively normalized upon LM11A-31 administration. The increased expression of PPARs and Nrf2 in RTT cells may represent a compensatory attempt to counteract OS and inflammation occurring in the pathology. As an instance, numerous experimental pieces of evidence demonstrate the direct role of PPARs and Nrf2 in upregulating the expression of the major antioxidant enzymes, such as [72], Gpx [73], SOD1 [74], and SOD2 [75]. However, the increased expression of these transcription factors in RTT fibroblasts was not accompanied by a concurrent induction of antioxidant enzymes. This apparent discrepancy may be explained by the fact that excessive OS elicits post-translational modifications, leading to the enhanced degradation of antioxidant enzymes. For instance, it has been reported that sustained ROS levels favor catalase ubiquitination and subsequent proteasomal degradation [76]. Similarly, other findings show that oxidative modifications promote SOD1 proteolysis [77].

In conclusion, this study further supports the involvement of OS and inflammation RTT pathology. Importantly, we provided evidence that the modulation of the p75NTR receptor by LM11A-31 can improve these alterations.

Using a modulator of p75NTR instead of molecules that directly target OS and inflammation may prove to be even more therapeutically beneficial. Indeed, signaling pathways mediated by p75NTR deeply influence numerous biological processes, particularly in the central nervous system, such as synaptic plasticity, energy metabolism, and neurodevelopment. Since all these processes are altered in RTT, the use of compounds like LM11A-31 could have the advantage of simultaneously targeting multiple aspects of the disease. Notably, LM11A-31 successfully entered clinical trials, showing a high safety profile and potential to counteract pathophysiological features in mild to moderate Alzheimer’s disease [78].

Despite the hints provided by this work, several aspects remain unclear and warrant further investigation. For instance, additional efforts are needed to better understand the intricate molecular mechanisms through which MeCP2 mutations can trigger OS and alterations in the enzymatic machinery controlling oxidative balance. Moreover, it is relevant to understand the signaling pathways involved in the effects mediated by LM11A-31. A detailed analysis of the molecular mechanisms could certainly help in interpreting the involvement of p75NTR in RTT and facilitate the identification of alternative approaches aimed at alleviating the symptomatology of the disease. A possible limitation of this study may rely on the small number of fibroblast donors employed in this study. However, it is relevant to note that redox imbalance and inflammation have already been observed in cultured fibroblasts derived from other RTT donors [12,13,79,80], emphasizing that these alterations are preserved characteristics regardless of the single affected individual. Moreover, given the multiorgan nature of RTT symptomatology, it would be beneficial to corroborate the findings of this study in diverse cellular models, particularly with respect to the impact on inflammatory mediators. Notably, despite the current acceptance that non-immune cells, such as fibroblasts, have immune properties and are capable of participating in immune processes by producing pro- and anti-inflammatory mediators [79], fibroblasts may not be fully immune-competent for the purposes of studying inflammation.

## Figures and Tables

**Figure 1 biomedicines-12-02624-f001:**
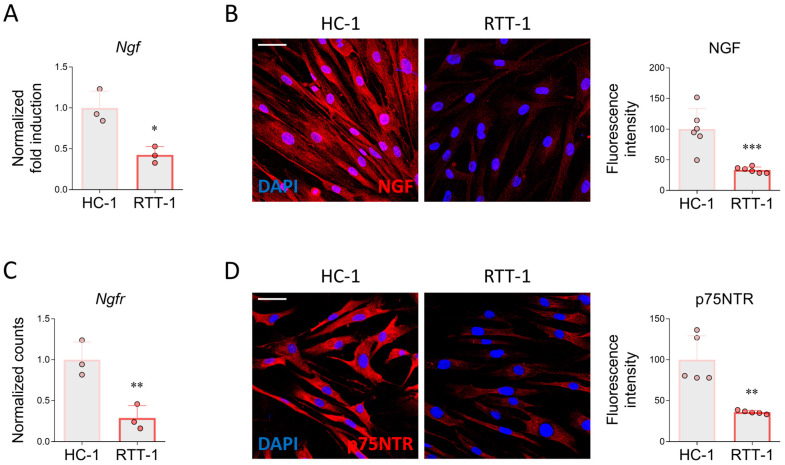
Expression of NGF and p75NTR fibroblasts derived from control individuals and Rett patients. (**A**) Total RNA was extracted from healthy control (HC-1) and Rett syndrome (RTT-1) fibroblasts, and the expression levels of *ngf* were measured by qRT-PCR. *n* = 3 biological replicates. Data represent means ± SD. (**B**) Immunofluorescence and respective quantitative analysis of NGF in HC and RTT. Cells were fixed in 4% PFA and stained with antibodies against NGF (red). DAPI (blue) was employed for nuclear counterstaining. *n* = 6 biological replicates. (**C**) qRT-PCR analysis of *ngfr* (*p75NTR*) in HC-1 and RTT-1 fibroblasts. *n* = 3 biological replicates. (**D**) Immunofluorescence and respective quantitative analysis of p75NTR immunoreactivity in HC-1 and RTT-1 cells. Cells were fixed in 4% PFA and stained with anti-p75NTR (red). DAPI (blue) was used to counterstain nuclei. *n* = 5 biological replicates. Data are expressed as mean ± SD. Statistical analysis was performed by using the Student’s unpaired *t*-test. Statistical significance is indicated as follows: * *p* < 0.05; ** *p* < 0.01; *** *p* < 0.001 vs. DMSO. Images were acquired using the Leica TCS SP8 confocal microscope and Leica Application Suite X (LAS X) software at 40× magnification. Scale bar: 50 µm.

**Figure 2 biomedicines-12-02624-f002:**
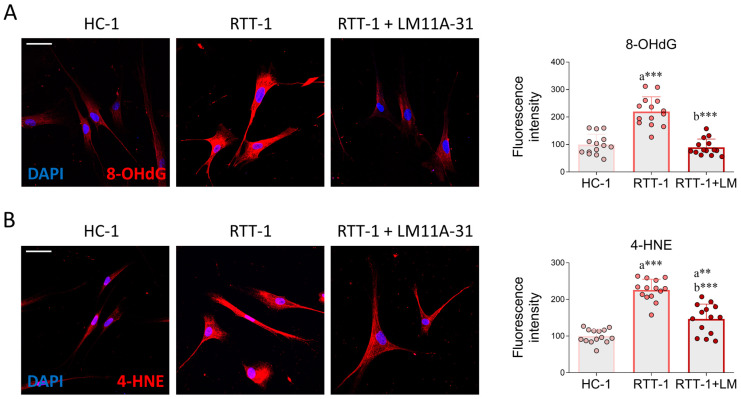
Effects of LM11A-31 on oxidative stress markers in RTT fibroblasts. Immunofluorescence and quantitative analysis of the OS biomarkers (**A**) 8-OHdG and (**B**) 4-HNE in healthy control fibroblast (HC-1), Rett syndrome fibroblasts (RTT-1), and RTT fibroblasts treated with LM11A-31 (RTT-1+LM) at the dose of 0.1 µM for 24 h. Cells were fixed in 4% PFA and stained with antibodies against 8-OHdG (red) or 4-HNE (red). Nuclei were counterstained with DAPI (blue). *n* = 14 biological replicates. Data are expressed as mean ± SD. Statistical analysis was performed using one-way ANOVA, followed by Tukey’s post hoc test. Statistical significance is indicated as follows: ** *p* < 0.01; *** *p* < 0.001. “a” indicates statistical significance vs. HC-1; “b” indicates statistical significance vs. RTT-1. Images were acquired using the Leica TCS SP8 confocal microscope and Leica Application Suite X (LAS X) software at 40× magnification. Scale bar: 50 µm.

**Figure 3 biomedicines-12-02624-f003:**
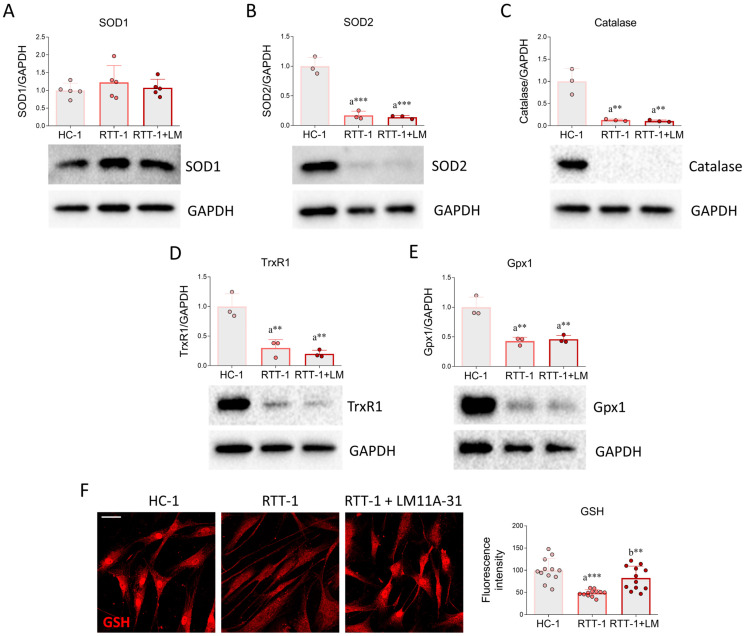
Effects of p75NTR modulation on the antioxidant response in RTT fibroblasts. Representative Western blot and densitometric analysis of (**A**) SOD1, (**B**) SOD2, (**C**) Catalase, (**D**) TrxR1, and (**E**) Gpx1 in healthy control fibroblasts (HC-1), Rett syndrome fibroblasts (RTT-1) and RTT fibroblasts treated with 0.1 μM of LM11A-31(RTT-1+LM) for 24 h. *n* = 3–5 biological replicates. GAPDH was used as a loading control. (**F**) Immunofluorescence and quantification of GSH immunoreactivity in HC-1, RTT-1, and RTT-1+LM experimental groups. Cells were fixed in 4% PFA and stained with antibodies against GSH (red). Nuclear staining was performed with DAPI (blue). *n* = 12 biological replicates. Data are expressed as mean ± SD. Statistical analysis was performed using one-way ANOVA, followed by Tukey’s post hoc test. Statistical significance is indicated as follows: ** *p* < 0.01; *** *p* < 0.001. “a” indicates statistical significance vs. HC-1; “b” indicates statistical significance vs. RTT-1. Images were acquired using the Leica TCS SP8 confocal microscope and Leica Application Suite X (LAS X) software at 40× magnification. Scale bar: 50 µm.

**Figure 4 biomedicines-12-02624-f004:**
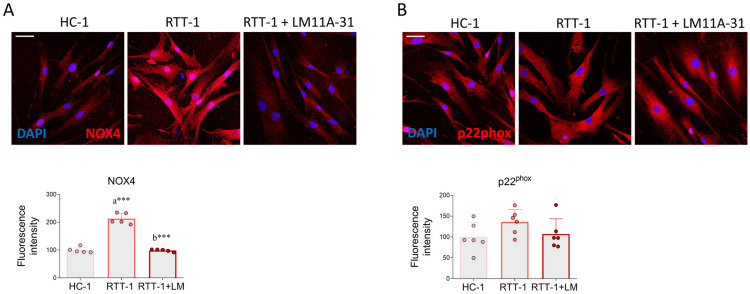
Impact of LM11A-31 on the expression of NADPH oxidase 4 subunits. Immunofluorescence and quantitative analysis of (**A**) NOX4 and (**B**) p22^phox^ fluorescence intensity in control fibroblast (HC-1), Rett syndrome fibroblasts (RTT-1), and RTT fibroblasts treated with LM11A-31 (0.1 μM for 24 h) (RTT-1+LM). Cells were fixed in 4% PFA and stained with antibodies against NOX4 (red) and p22^phox^. DAPI was employed for nuclear counterstaining. *n* = 5–6 biological replicates. Images were acquired using the Leica TCS SP8 confocal microscope and Leica Application Suite X (LAS X) software at 40× magnification. Scale bar: 50 µm. Data are expressed as mean ± SD. Statistical analysis was performed using a one-way ANOVA, followed by Tukey’s post hoc test. Statistical significance is indicated as follows: *** *p* < 0.001. “a” indicates statistical significance vs. HC-1; “b” indicates statistical significance vs. RTT-1.

**Figure 5 biomedicines-12-02624-f005:**
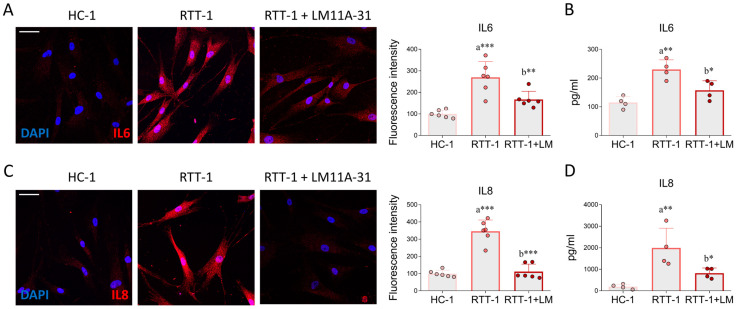
p75NTR modulation reduces IL-6 and IL-8 expression in RTT fibroblasts. Representative immunofluorescence and respective quantitative analysis of (**A**) IL-6 immunoreactivity in control fibroblasts (HC-1), Rett syndrome fibroblasts (RTT-1), and RTT fibroblasts treated with LM11A-31(RTT-1+LM) at the dose of 0.1 μM for 24 h. Cells were fixed in 4% PFA and stained with antibodies against IL-6 (red). DAPI was used to counterstain nuclei. *n* = 6 biological replicates. Images were acquired using the Leica TCS SP8 confocal microscope and Leica Application Suite X (LAS X) software at 40× magnification. Scale bar: 50 µm. (**B**) ELISA on IL-6 in culture medium from HC-1, RTT-1, and RTT-1+LM groups treated as abovementioned. *n* = 4 biological replicates. (**C**) IL-8 immunoreactivity (red) and respective quantitative analysis performed on HC-1, RTT-1, and RTT-1+LM fibroblasts treated as in (**A**). DAPI was used for nuclear staining. *n* = 6 biological replicates. Scale bar: 50 µm. (**D**) ELISA on secreted IL-8 in conditioned medium from HC-1, RTT-1, and RTT-1+LM cells treated as abovementioned. *n* = 4 biological replicates. Data are expressed as mean ± SD. Statistical analysis was performed using one-way ANOVA, followed by Tukey’s post hoc test. Statistical significance is indicated as follows: * *p* < 0.05; ** *p* < 0.01; *** *p* < 0.001. “a” indicates statistical significance vs. HC-1; “b” indicates statistical significance vs. RTT-1.

**Figure 6 biomedicines-12-02624-f006:**
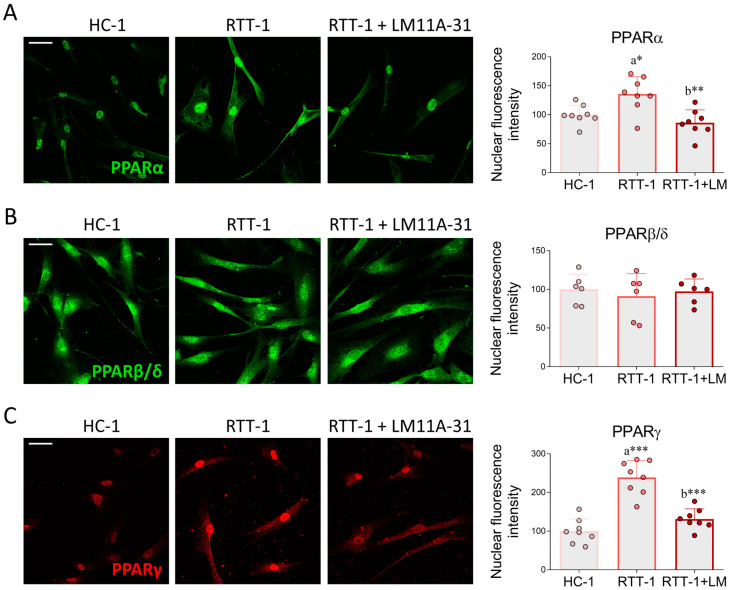
LM11A-31 modulates PPAR expression in RTT fibroblasts. Immunofluorescence and signal intensity analysis of (**A**) PPARα, (**B**) PPARβ/δ, and (**C**) PPARγ in control fibroblast (HC-1), Rett syndrome fibroblasts (RTT-1) and RTT fibroblasts treated with 0.1 μM of LM11A-31 (RTT-1+LM) for 24 h. Cells were fixed in 4% PFA and stained with antibodies against PPARα (green), PPARβ/δ (green) and PPARγ (red). Nuclei were counterstained with DAPI (blue). *n* = 6–8 biological replicates. Data are expressed as mean ± SD. Statistical analysis was performed using a one-way ANOVA, followed by Tukey’s post hoc test. Statistical significance is indicated as follows: * *p* < 0.05; ** *p* < 0.01; *** *p* < 0.001. “a” indicates statistical significance vs. HC-1; “b” indicates statistical significance vs. RTT-1. Images were acquired using the Leica TCS SP8 confocal microscope and Leica Application Suite X (LAS X) software at 40× magnification. Scale bar: 50 µm.

**Figure 7 biomedicines-12-02624-f007:**
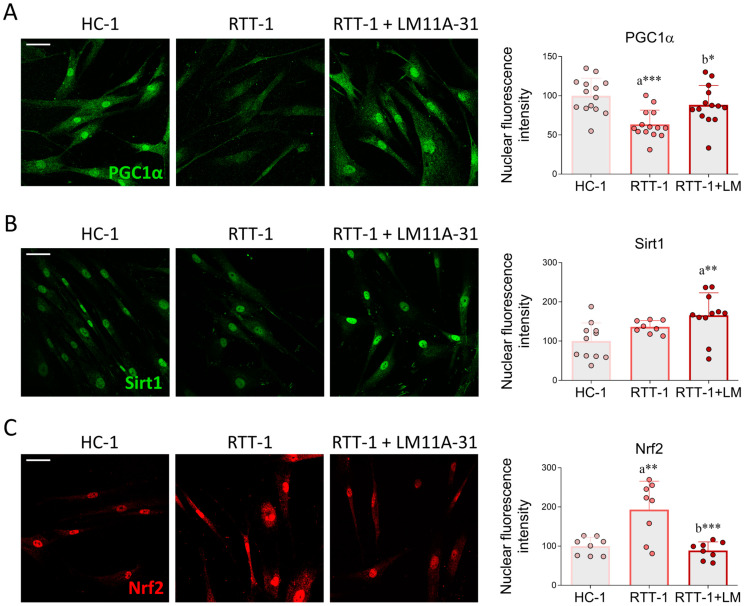
LM11A-31 influences the expression of transcription factors involved in redox homeostasis. Immunofluorescence and quantification of signal intensity of (**A**) PGC1α, (**B**) Sirt1, and (**C**) Nrf2 in control fibroblast (HC-1), Rett syndrome fibroblasts (RTT-1), and RTT fibroblasts treated with LM11A-31 (RTT-1+LM) at the dose of 0.1 μM for 24 h. Cells were fixed in 4% PFA and stained with antibodies against PGC1α (green), Sirt1 (green) and Nrf2 (red). DAPI was employed for nuclear counterstaining. *n* = 8–14 biological replicates. Data are expressed as mean ± SD. Statistical analysis was performed using a one-way ANOVA, followed by Tukey’s post hoc test. Statistical significance is indicated as follows: * *p* < 0.05; ** *p* < 0.01; *** *p* < 0.001. “a” indicates statistical significance vs. HC-1; “b” indicates statistical significance vs. RTT-1. Images were acquired using the Leica TCS SP8 confocal microscope and Leica Application Suite X (LAS X) software at 40× magnification. Scale bar: 50 µm.

## Data Availability

The original contributions presented in the study are included in the article/Appendix A; further inquiries can be directed to the corresponding author.

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
