# Peer review of "p75NTR Modulation Reduces Oxidative Stress and the Expression of Pro-Inflammatory Mediators in a Cell Model of Rett Syndrome"

_biomedicines, 2024, doi:10.3390/biomedicines12112624_

Round 1
Reviewer 1 Report
Comments and Suggestions for Authors
This study offers valuable insights into targeting oxidative stress and inflammation in Rett syndrome (RTT) through p75NTR modulation using LM11A-31, presenting a novel therapeutic approach. However, some sections need refinement.
The introduction is overly detailed and could be streamlined, moving certain explanations to the discussion for clarity.
Additional methodological details, such as sample sizes, control selection, and antibody validation, would strengthen reproducibility, especially by providing demographic information on fibroblast donors and validating immunofluorescence antibodies.
Consistent comparisons with healthy controls in results and enhanced figure contrast would clarify LM11A-31’s effects.
Mechanistically, further discussion on how p75NTR modulation may counter oxidative and inflammatory dysregulation specific to RTT would add depth, as would addressing limitations of fibroblast models for neurodevelopmental conditions.
Minor revisions include, improving graphical clarity, and correcting minor typographical issues such as in "experimental model fibroblasts."
Additionally, certain phrases (e.g., “blunting oxidative stress”) could be rephrased for scientific precision. Consider using “modulation” instead of “blunting” to describe LM11A-31’s effects on oxidative markers to enhance readability and precision
These revisions would improve the manuscript's rigor and clarity.
Author Response
- Comments 1: “This study offers valuable insights into targeting oxidative stress and inflammation in Rett syndrome (RTT) through p75NTR modulation using LM11A-31, presenting a novel therapeutic approach. However, some sections need refinement”
Response 1: We would like to thank the reviewer for the appreciation of our work and for the helpful comments that allowed us to improve the manuscript.
- Comments 2: “The introduction is overly detailed and could be streamlined, moving certain explanations to the discussion for clarity.”
Response 2 : We thank the reviewer for this observation and agree that the Introduction section contains excessively detailed information that can be streamlined. In the new version, we have simplified the details of the Introduction, many of which were already covered in the Discussion section. Where appropriate, we have moved some explanations to the Discussion section as suggested.
- Comments 3: “Additional methodological details, such as sample sizes, control selection, and antibody validation, would strengthen reproducibility, especially by providing demographic information on fibroblast donors and validating immunofluorescence antibodies.”
Response 3: According to the reviewer’s suggestion, we added more methodological details about the validation of antibodies for immunofluorescence (please, see paragraph 2.4 : «All primary antibodies employed in this study are commercially available and were selected based on their published validation. The anti-4-HNE, anti-8-OHdG and anti-Nrf2 antibodies were subjected to further testing in our laboratory on different human cell lines (HepG2, SH-SY5Y and fibroblasts) treated with oxidative stress inducers (300 µM H2O2, 0.1 µM rotenone) as positive controls. The anti-IL-6 and anti-IL-8 antibodies were validated using LPS-stimulated THP-1 cells as a positive control. The anti-NGF and anti-p75NTR antibodies were validated using blocking peptides »).
The fibroblasts employed in this study have been previously utilized in a published work. Therefore, in order to avoid unnecessary repetition, we refer the interested reader to our previous publication (doi:10.1371/journal.pone.0104834) for details regarding demographic information, subject selection and ethical statements. These details can be found in the manuscript.
- Comments 4: “Consistent comparisons with healthy controls in results and enhanced figure contrast would clarify LM11A-31’s effects”
Response 4: In the “Results” Section we now compared the effect of LM11A-31 not only in relation to Rett cells, but also in relation to healthy controls. Where possible, we have increased the contrast of the images to better visualize the effects induced by LM11A-31. Nevertheless, we refrained from excessively emphasizing the contrast in the images to prevent any undue distortion of the original images.
- Comments 5: “Mechanistically, further discussion on how p75NTR modulation may counter oxidative and inflammatory dysregulation specific to RTT would add depth, as would addressing limitations of fibroblast models for neurodevelopmental conditions.”
Response 5: We thank the reviewer for these suggestions, which gave us the opportunity to propose some molecular explanations linking p75NTR to inflammation and/or oxidative stress. Identifying the transduction pathways activated by p75NTR is quite challenging, as this receptor can induce diverse signalling cascades depending on the cellular context and physiopathological condition, due to its ability to interact with a plethora of adaptor proteins. For this reason, in the previous version of the manuscript, we preferred to be cautious in our interpretations. In the new version, we have discussed the possible molecular mechanisms involved in the effects reported in this paper (please see “Discussion” Section: “The role of p75NTR has been already associated to the modulation of the antioxidant machinery: p75NTR-negative cells are less resilient to oxidative damage in PC12 cells, supporting the notion that reduced p75NTR expression may also affect redox homeo-stasis in RTT cells. Notably, the intracellular domain of p75NTR is an essential prerequisite for regulating glutathione oxidation and protecting PC12 cells against oxidative stress. Despite this evidence, the molecular pathways involved in the antioxidant effects of p75NTR need to be further investigated. The intricate nature of the signal transduction cascades associated with p75NTR presents a fascinating challenge to gain deeper insights into its biological functions.” […] “To date, the molecular mechanisms linking p75NTR activity and the modulation of NADPH oxidase subunits are still unexplored and deserve further investigation. Nevertheless, elegant work has demonstrated that p75NTR loss markedly diminishes the expression of Siah2, a ubiquitin ligase responsible for HIF-1α degradation. Consequently, the absence of p75NTR promotes the stabilization of HIF-1. As nox4 gene is under the transcriptional control of HIF-1α, it is possible to speculate that the reduction of p75NTR observed in RTT fibroblasts facilitates the upregulation of the NOX4 subunit through the contribution of HIF-1α.” […] Although the downstream pathways of p75NTR leading to modulation of inflammation are still unclear, it has been shown that this receptor can recruit signalling proteins such as interleukin receptor associated kinase (IRAK), which are generally recruited by interleukin-1 receptor (IL1R) or toll-like receptors (TLRs) to activate NF-kB and promote a pro-inflammatory response. Thus, it has been postulated that p75NTR may sequester IRAK that is no longer available to compete with receptors responsible for initiating the inflammatory cascade. Supporting this hypothesis, other findings show that p75NTR modulation by LM11A-31 suppresses NF-kB activation in diverse physiopathological contexts”).
Concerning the use of fibroblasts as a model for neurodevelopmental conditions, as already reported in the “Introduction” section, Rett syndrome is no longer exclusively considered as a neurodevelopmental disorder, because of the presence of several multiorgan dysfunctions. In this scenario, peripheral cells can effectively be employed as experimental models to study specific aspects of the disease (doi: 10.1016/j.ddmod.2019.11.001). In particular, it is now clear that oxidative stress is a systemic burden in Rett syndrome, affecting different cells/tissues such as blood cells, fibroblasts and brain cells (doi: 10.1016/j.neubiorev.2018.12.009). However, we adhere to the reviewer's considerations, especially in the context of assessing inflammation in non-fully immune-competent cells. In this regard, we have included a section dedicated to the possible limitations of the study (please see “Discussion” section: “Moreover, given the multiorgan nature of RTT symptomatology, it would be beneficial to corroborate the findings of this study in diverse cellular models, particularly with respect to the impact on inflammatory mediators. Notably, despite the current acceptance that non-immune cells, such as fibroblasts, have immune properties and are capable of participating in immune processes by producing pro- and an-ti-inflammatory mediators, fibroblasts may not be fully immune competent for the purposes of studying inflammation.”).
- Comments 6: “Minor revisions include, improving graphical clarity, and correcting minor typographical issues such as in "experimental model fibroblasts.”
Response 6: We thank the reviewer for the suggestions. The manuscript was accurately revised to improve graphical clarity and eliminate possible typographical errors.
- Comments 7: “Additionally, certain phrases (e.g., “blunting oxidative stress”) could be rephrased for scientific precision. Consider using “modulation” instead of “blunting” to describe LM11A-31’s effects on oxidative markers to enhance readability and precision”
Response 7: Following the reviewer's suggestion, the new version of the manuscript avoids the use of "blunting" and replaces it with "reducing", as LM11A-31 actually reduces oxidative stress as shown by the decreased levels of oxidative damage markers.

Reviewer 2 Report
Comments and Suggestions for Authors
The Authors described their research in detail and presented the results obtained in great detail. The introduction lacks some important information in some places, which if added will contribute to a better understanding of the topic.
Introduction - it should be mentioned at what age the first symptoms appear
Line 125 - it may be that the same cell cultures have already been used, but brief information on at least the age and sex of the donors should also be inserted in this manuscript
Author Response
- Comments 1: “The Authors described their research in detail and presented the results obtained in great detail. The introduction lacks some important information in some places, which if added will contribute to a better understanding of the topic.
Introduction - it should be mentioned at what age the first symptoms appear”
Response 1: We would like to thank the reviewer for appreciating our work and for the helpful comments.
In accordance with the comments of reviewer 1, who highlighted the excessive detail in the Introduction, the section has been remodeled by removing superfluous information, which has been incorporated into the “Discussion” section. Nevertheless, we agree with this reviewer that it is important to mention the onset of the pathology, and thus the introduction has been modified as follows: “Most of the patients with RTT develop normally up to 6-18 months of age [1]. Subsequently, they present with a typical clinical picture that includes four stages of neurological regression, summarized by the loss of cognitive, social and motor skills acquired in early life [4,5].”
- Comments 2: “Line 125 - it may be that the same cell cultures have already been used, but brief information on at least the age and sex of the donors should also be inserted in this manuscript”
Response 2: As assessed by the reviewer, the fibroblasts employed in this study have been previously utilized in a published work. Therefore, in order to avoid unnecessary repetition, we refer the interested reader to our previous publication (doi:10.1371/journal.pone.0104834) for details regarding demographic information, subject selection and ethical statements. However, according to the reviewer’s request, we included in our manuscript details about sex and mean age of the enrolled patients.

Reviewer 3 Report
Comments and Suggestions for Authors
In the study “p75NTR modulation blunts oxidative stress and reduces the expression of pro-inflammatory mediators in a cell model of Rett syndrome“ by Varone et al., the authors explore the effects of LM11A-31 mediated p75NTR modulation on oxidative stress and inflammation in the fibroblasts from Rett patients. They found that p75NTR modulation normalizes the expression levels of transcription factors involved in the regulation of antioxidant response and inflammation.
The authors state that all the experiments were done on two healthy controls and two Rett patients’ fibroblasts. How is the use of only two subjects justified for further statistical analyses?
In the chapter 3.1 NGF and p75NTR expression is reduced in RTT fibroblasts, the authors state that the expression of ngf in RTT fibroblasts is reduced. They present three different experiments. Considering that fibroblasts were isolated from 2 subjects it is unclear how this experiment was done and where is the statistical difference coming from. Were the cells from two Rett patients or two healthy controls pooled together and then plated and treated in three/six different experimental replicates? If so, how is it justified to pool the cells from different human subjects? The same question stands for all the subsequent experiments. In addition, what is the difference between different experiments (Line 255) and experimental replicates (Line 256). Aren’t all the experiments presented in the study experimental replicates? The question rises if these experimental replicates are from one human subject or from both (pooled samples) per group?
Line 465: Any hypothesis on how the alterations in p75NTR signaling pathway may significantly contribute to derangements of the redox balance in RTT.
Line 519: Some explanation on how the p75NTR modulation is affecting pro-inflammatory pathways will be beneficial for the readers.
Line 545: The authors state that LM11A-31 successfully entered clinical trials, showing a high safety profile and potential to counteract neurological disorders but did not provide any reference. Are these clinical trials in relation to the Rett syndrome? If LM11A-31 have already entered clinical trials than the specific contribution in terms of novelty regarding LM11A-31 effects explored in this study should be emphasized.
Author Response
- Comments 1: “The authors state that all the experiments were done on two healthy controls and two Rett patients’ fibroblasts. How is the use of only two subjects justified for further statistical analyses?
In the chapter 3.1 NGF and p75NTR expression is reduced in RTT fibroblasts, the authors state that the expression of ngf in RTT fibroblasts is reduced. They present three different experiments. Considering that fibroblasts were isolated from 2 subjects it is unclear how this experiment was done and where is the statistical difference coming from. Were the cells from two Rett patients or two healthy controls pooled together and then plated and treated in three/six different experimental replicates? If so, how is it justified to pool the cells from different human subjects? The same question stands for all the subsequent experiments. In addition, what is the difference between different experiments (Line 255) and experimental replicates (Line 256). Aren’t all the experiments presented in the study experimental replicates? The question rises if these experimental replicates are from one human subject or from both (pooled samples) per group?”
Response 1: We would like to thank the reviewer for these comments, which allow us to better clarify the sample size indicated in the experiments and the nature of the cells used. In this study, we employed cultured fibroblasts derived from two healthy individuals and two patients with Rett syndrome.
Specifically, all experiments shown in the main figures are from biological replicates derived from the cells of a single healthy individual and a single Rett patient. This means that the experiments were reproduced at least in biological triplicate on different sets of cells, but derived from the same healthy control and the same Rett patient. In other words, the cells of the individuals were not pooled. To facilitate the interpretation of the data, cells derived from different healthy subjects and Rett patients are now marked as HC-1, HC-2, RTT-1, RTT-2. In the supplementary figures, results obtained from cells derived from a second healthy individual and a second Rett patient have been included, with the aim of demonstrating that the observed modulations are independent of the specific individual's background. This approach is frequently used in studies using fibroblasts derived from patients suffering from rare diseases, whose availability may be limited (doi: 10.1016/j.jlr.2021.100114; doi: 10.1089/adt.2009.0240; doi: 10.1101/2023.12.19.572384; doi: 10.1074/mcp.M114.045609; doi: 10.1523/JNEUROSCI.3735-16.2017; doi: 10.1016/j.scr.2017.02.017; doi: 10.1083/jcb.201903018; doi: 10.15252/emmm.201910270; doi: 10.3389/fcell.2020.610427). Additionally, the isolation of fibroblasts derives from skin biopsy, which is considered an invasive procedure and not easy to have access (doi: 10.1016/j.abb.2023.109860).
Importantly, increased oxidative stress, together with the presence of a pro-inflammatory state, has already been reported not only in animal models of Rett syndrome, but also in other studies performed on fibroblasts isolated from other Rett patients, emphasizing that oxidative stress and inflammation are preserved characteristics regardless of the single affected individual, so that cells isolated from 2 subjects can be considered adequate prototypes (doi: 10.1016/j.redox.2019.101334; doi: 10.1016/j.abb.2020.108660; doi: 10.1155/2014/195935; doi: 10.1016/j.bbadis.2015.07.014; doi: 10.1186/s12967-023-04622-5). In any case, we are aware of the reviewer's considerations and the low number of patients recruited has been discussed as a potential limitation of the study (please see “Discussion” section: “A possible limitation of this study may rely on the small number of fibroblast donors employed in this study. However, it is relevant to note that redox imbalance and inflammation have been already observed in cultured fibroblasts derived from other RTT donors, emphasizing that these alterations are preserved characteristics regardless of the single affected individual”).
Regarding the use of the terms "different replicates" and "experimental replicates", there is no difference in meaning. However, we realize that these definitions may raise doubts of interpretation in the reviewer as well as in the readers, therefore the terms have been replaced in the text with the term "biological replicates". The term "biological replicate" indicates that each experimental point shown in the graph comes from distinct samples. Thus, each dot on the graph corresponds to an independent experimental set and does not represent a technical replicate (the same sample repeatedly measured several times).
- Comments 2: “Line 465: Any hypothesis on how the alterations in p75NTR signaling pathway may significantly contribute to derangements of the redox balance in RTT. Line 519: Some explanation on how the p75NTR modulation is affecting pro-inflammatory pathways will be beneficial for the readers.”
Response 2: We are grateful to the reviewer for this suggestion, which provided us with the opportunity to propose potential molecular explanations linking p75NTR to oxidative stress and inflammation, especially in the context of our results. Identifying the transduction pathways activated by p75NTR is challenging, as this receptor can induce diverse signalling cascades depending on the cellular context and physiopathological condition, due to its ability to interact with a plethora of adaptor proteins. Consequently, in the original version of the manuscript, we preferred to be cautious in our interpretations. In the new version, we have discussed the possible molecular mechanisms involved in the effects reported in this paper (please see the "Discussion" section).: “The role of p75NTR has been already associated to the modulation of the antioxidant machinery: p75NTR-negative cells are less resilient to oxidative damage in PC12 cells, supporting the notion that reduced p75NTR expression may also affect redox homeo-stasis in RTT cells. Notably, the intracellular domain of p75NTR is an essential prerequisite for regulating glutathione oxidation and protecting PC12 cells against oxidative stress. Despite this evidence, the molecular pathways involved in the antioxidant effects of p75NTR need to be further investigated. The intricate nature of the signal transduction cascades associated with p75NTR presents a fascinating challenge to gain deeper insights into its biological functions.” […] “To date, the molecular mechanisms linking p75NTR activity and the modulation of NADPH oxidase subunits are still unexplored and deserve further investigation. Nevertheless, elegant work has demonstrated that p75NTR loss markedly diminishes the expression of Siah2, a ubiquitin ligase responsible for HIF-1α degradation. Consequently, the absence of p75NTR promotes the stabilization of HIF-1. As nox4 gene is under the transcriptional control of HIF-1α, it is possible to speculate that the reduction of p75NTR observed in RTT fibroblasts facilitates the upregulation of the NOX4 subunit through the contribution of HIF-1α.” […] Although the downstream pathways of p75NTR leading to modulation of inflammation are still unclear, it has been shown that this receptor can recruit signalling proteins such as interleukin receptor associated kinase (IRAK), which are generally recruited by interleukin-1 receptor (IL1R) or toll-like receptors (TLRs) to activate NF-kB and promote a pro-inflammatory response. Thus, it has been postulated that p75NTR may sequester IRAK that is no longer available to compete with receptors responsible for initiating the inflammatory cascade. Supporting this hypothesis, other findings show that p75NTR modulation by LM11A-31 suppresses NF-kB activation in diverse physiopathological contexts”).
- Comments 3: “Line 545: The authors state that LM11A-31 successfully entered clinical trials, showing a high safety profile and potential to counteract neurological disorders but did not provide any reference. Are these clinical trials in relation to the Rett syndrome? If LM11A-31 have already entered clinical trials than the specific contribution in terms of novelty regarding LM11A-31 effects explored in this study should be emphasized.”
Response 3: We thank the reviewer for this comment, which provides us with the opportunity to more clearly elucidate the clinical information related to LM11A-31. It is noteworthy that LM11A-31 has never been tested in the context of Rett syndrome. However, this molecule is attracting interest due to its successful achievement of primary endpoints related to safety and tolerability in a double-blind phase 2a clinical trial. Furthermore, with regard to the prespecified secondary and exploratory outcomes, LM11A-31 has demonstrated efficacy in slowing the progression of pathophysiological features associated with Alzheimer's disease (AD). Results about this clinical trial were recently published (doi: 10.1038/s41591-024-02977-w). Details are now specified in the “Introduction” section and the reference has been provided.

Reviewer 4 Report
Comments and Suggestions for Authors
This study explores the effects of LM11A-31, a p75 neurotrophin receptor modulator, on fibroblasts from Rett syndrome (RTT) patients. The treatment reduced oxidative stress markers, decreased pro-oxidant enzyme NOX4, and lowered inflammatory cytokines IL-6 and IL-8. Additionally, it normalized transcription factors related to antioxidant responses. These results suggest that p75NTR modulation may be a potential therapeutic target to improve redox balance and reduce inflammation in RTT. Generally, the quality of manuscript is quite good except a few minor issues.
Comments:
1. Could authors provide more details about how the microscopy images were processed and quantified in the method section? (e.g. How many replicates for each condition, and how many images for each sample)?
2. Why did this study use fibroblasts not other cell lines as model?
3. For LM11A-137 treatment, why did authors use 100nM in the final culture? Is this close to actual physiological condition in human body?
4. Please add scale bars for all images.
Author Response
- Comments 1: “This study explores the effects of LM11A-31, a p75 neurotrophin receptor modulator, on fibroblasts from Rett syndrome (RTT) patients. The treatment reduced oxidative stress markers, decreased pro-oxidant enzyme NOX4, and lowered inflammatory cytokines IL-6 and IL-8. Additionally, it normalized transcription factors related to antioxidant responses. These results suggest that p75NTR modulation may be a potential therapeutic target to improve redox balance and reduce inflammation in RTT. Generally, the quality of manuscript is quite good except a few minor issues.”
Response 1: We would like to thank the reviewer for the positive assessment of our work and for the constructive feedback, which has enabled us to enhance the manuscript.
- Comments 2: “Could authors provide more details about how the microscopy images were processed and quantified in the method section? (e.g. How many replicates for each condition, and how many images for each sample)?”
Response 2: According to the reviewer’s observation, details about the number of replicates are included in the figure legends. To avoid doubts of interpretation in the readers, we specified the replicates for each condition as "biological replicates". The term "biological replicate" indicates that each experimental point shown in the graph comes from distinct samples. Thus, each dot on the graph corresponds to an independent experimental set and does not represent a technical replicate (the same sample repeatedly measured several times). Additional information about processing and quantification of microscopy images is now included in the paragraph 2.4 (“Signal quantification was calculated as the mean fluorescence intensity per cell area by using ImageJ v1.54d software (National Institutes of Health, Bethesda, MD, USA) for Windows 10, according to the previously reported procedure (Salzano et al., 2024 ). In order to ensure the absence of operator biases, the results concerning immunofluorescence quantification were additionally confirmed through the automated analysis of the same images with the software Q-IF (Salzano et al., 2024 ). The interpretation of the quantitative analysis is now better specified in the “Materials and Methods Section” (please see paragraph 2.4: “The dots dispersed around the SD represent individual values, each derived from the average fluorescence of the cells analyzed in a single image (each image is from a different experiment”).
- Comments 3: “Why did this study use fibroblasts not other cell lines as model?”
Response 3: Primary fibroblasts are regarded as a valuable disease-in-a-dish model for the study of genetic diseases, including Rett syndrome. Notably, all cells of the human body carry the Mecp2 gene mutation, irrespective of the phenomenon of X chromosome inactivation. Cells isolated from individuals with Rett syndrome, with particular reference to skin fibroblasts, may represent an optimal model for investigating the molecular mechanisms linking the Mecp2 mutation to the Rett syndrome phenotype. As previously demonstrated in multiple reports, cultured fibroblasts derived from Rett syndrome patients accurately reflect the events occurring in the systemic compartment and are frequently employed to assess the efficacy of novel pharmacological strategies (doi: 10.1016/j.ddmod.2019.11.001). A brief explanation regarding the selection of the experimental model has now been included in the Introduction section (“Fibroblasts from RTT individuals can be considered valuable in vitro models of the disease, as they adequately mirror the systemic oxidative stress and inflammation found in RTT patients”).
- Comments 4: “For LM11A-137 treatment, why did authors use 100nM in the final culture? Is this close to actual physiological condition in human body?”
Response 4: We thank the reviewer for this consideration, which give us the chance to better explain the selection of the dose for LM11A-31 treatments. We used the dose of 100 nM as this is the most employed for in vitro studies (doi: 10.3389/ebm.2024.10123; doi: 10.1038/s41598-020-77210-y; doi: 10.1371/journal.pone.0003604). This information is now reported in the “Materials and Methods” Section (“The working solution was then added to the cultured cells (1:1000 dilution) to give a final concentration of 100 nM of LM11A-31, a dose generally employed in cell culture studies”). It is notable that the concentration of 100 nM is below the plasma concentration of the drug observed in vivo and demonstrated to be efficacious in counteracting neuronal degeneration in mice (doi: 10.1016/j.neurobiolaging.2013.02.015). To the best of our efforts, even though LM11A-31 successfully passed phase II clinical trial, we were not able to find any published data about pharmacokinetics in humans.
- Comments 5: “Please add scale bars for all images.”
Response 5: We extend our apologies for this inadvertent omission. In accordance with the reviewer's request, the scale bars have now been integrated into the images.

Round 2
Reviewer 1 Report
Comments and Suggestions for Authors
I went through the revised version of the manuscript. The authors addressed all the issues raised and the manuscript is greatly improved and can be published in its present form.
Author Response
- Comments 1: “I went through the revised version of the manuscript. The authors addressed all the issues raised and the manuscript is greatly improved and can be published in its present form.”
Response 1: We would like to express our gratitude once more to the reviewer for the valuable feedback, which has enabled us to enhance our manuscript and for their positive assessment of our response to the raised concerns.

Reviewer 3 Report
Comments and Suggestions for Authors
I think that the authors have adequately addressed the comments made by the reviewers in the revised version of the manuscript. Therefore, I have no further comments.
Author Response
- Comments 1: “I think that the authors have adequately addressed the comments made by the reviewers in the revised version of the manuscript. Therefore, I have no further comments.”
Response 1: We thank the reviewer again for his/her comments, which have helped us to improve our manuscript, and for his appreciation of the way in which we have addressed the concerns raised.
